# Revisiting Motion Information for RGB-Event Tracking with MOT Philosophy

**Tianlu Zhang**
EMIM
Xidian University
tlzhang96@outlook.com

**Kurt Debattista**
Warwick Manufacturing Group
University of Warwick
K.Debattista@warwick.ac.uk

**Qiang Zhang***
EMIM
Xidian University
qzhang@@xidian.edu.cn

**Guiguang Ding**
School of Software
Tsinghua University
dinggg@tsinghua.edu.cn

**Jungong Han***
Department of Automation
Tsinghua University
jungonghan77@gmail.com

## Abstract

RGB-Event single object tracking (SOT) aims to leverage the merits of RGB and event data to achieve higher performance. However, existing frameworks focus on exploring complementary appearance information within multi-modal data, and struggle to address the association problem of targets and distractors in the temporal domain using motion information from the event stream. In this paper, we introduce the Multi-Object Tracking (MOT) philosophy into RGB-E SOT to keep track of targets as well as distractors by using both RGB and event data, thereby improving the robustness of the tracker. Specifically, an appearance model is employed to predict the initial candidates. Subsequently, the initially predicted tracking results, in combination with the RGB-E features, are encoded into appearance and motion embeddings, respectively. Furthermore, a Spatial-Temporal Transformer Encoder is proposed to model the spatial-temporal relationships and learn discriminative features for each candidate through guidance of the appearance-motion embeddings. Simultaneously, a Dual-Branch Transformer Decoder is designed to adopt such motion and appearance information for candidate matching, thus distinguishing between targets and distractors. The proposed method is evaluated on multiple benchmark datasets and achieves state-of-the-art performance on all the datasets tested.

## 1   Introduction

Single object tracking (SOT) aims to predict the position of a target in videos, by being given only the position of the target in the initial frame. While traditional RGB-based trackers [1, 8] can effectively capture comprehensive scene representations, including color and semantic information, they face significant performance degradation in challenging conditions like fast motion, low illumination and distractions from similar objects.

To address such challenges associated with frame-based cameras, some researchers have taken advantage of event cameras [39], which are characterized by high temporal resolution and high dynamic range, to augment the RGB data for reliable object tracking. In the past few years, various methods have been proposed for RGB-Event (RGB-E) object tracking. Existing RGB-E trackers primarily concentrate on exploring complementary appearance information within RGB and event

---

*Corresponding author.

38th Conference on Neural Information Processing Systems (NeurIPS 2024).

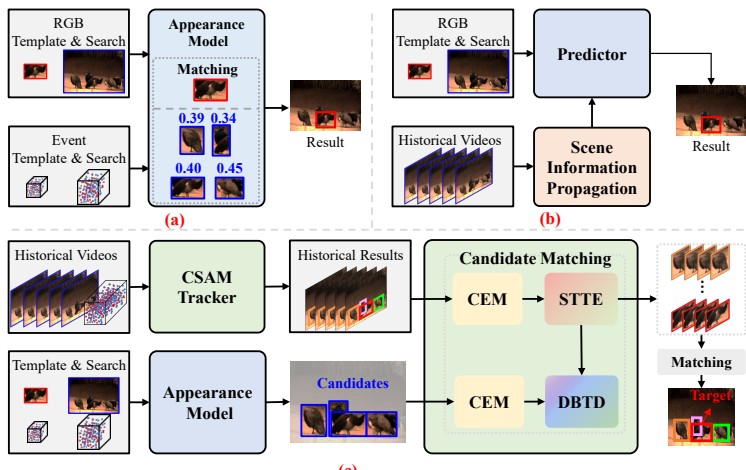

Figure 1: Architectures of different RGB-E tracking frameworks. (a) RGB-E tracker based on appearance information. (b) RGB tracker based on scene information propagation. (c) Our proposed CSAM framework.

data to enhance tracking performance with three typical approaches i.e., early fusion method [29], middle fusion method [37, 38] and one-stream method [24, 44]. Despite achieving commendable improvements, mainstream RGB-E tracking algorithms still cannot solve the association problem of the targets and distractor objects in the temporal domain, as shown in Fig. 1 (a).

Alternatively, some RGB trackers [2, 3, 19] propagate valuable scene information through the sequence to improve their discriminative ability, as shown in Fig. 1 (b). These methods mine the information in two main ways. The first one relies on implicitly transforming the scene information to locate the targets, and the scene information is generally represented using a set of embeddings [2, 3]. To ensure effective transformation and avoid introducing noisy information, these methods usually require careful design of the encoding strategy to obtain effective scene information embeddings. The second approach explicitly explores the scene information by simultaneously keeping track of both targets and distractors [19, 41]. However, these methods are susceptible to environmental interference, and their matching strategies relying on appearance information may miss the target when the target and distractor trajectories are close.

In fact, event data can not only provide the edge information to improve the RGB feature representations but also contains abundant motion cues to reflect the motion state of the objects, which is meaningful to differentiate between targets and distractors, even if they may look similar. Motivated by these observations, we propose an Appearance-Motion Modeling RGB-E tracking framework with a Cascade Structure, referred to as CSAM, that goes beyond leveraging complementary appearance information within RGB-E data. As shown in Fig. 1 (c), the proposed CSAM framework employs an appearance model to initially determine the candidates with similar appearance to the targets, and then designs a candidate matching network with encoder-decoder structure to dynamically incorporate motion information contained in the RGB-E videos to track all the candidates with the Multi-Object Tracking (MOT) philosophy. The candidate that matches the historical target tracklet will be regarded as the final tracking result.

Specifically, the candidate matching network consists of a Candidate Encoding Module (CEM), a Spatial-Temporal Transformer Encoder (STTE), and a Dual-branch Transformer Decoder (DBTD). Recognizing the critical roles of the two types of information in candidate association, our proposed CEM is used to encode both appearance cues and motion representations for each candidate. Subsequently, an STTE block, comprising a Spatial Encoder and a Temporal Encoder, is introduced to model spatial-temporal relationships among candidates by synergistically utilizing such appearance and motion embeddings. Finally, a DBTD block, comprising a Spatial-temporal Decoder and a Motion Decoder, is presented to match candidates with historical tracklets by using both appearance and motion information.

**Contributions:** In summary, our contributions are: **(i)** We propose a novel RGB-E tracking framework, i.e., CSAM, which first predicts the candidates by using an appearance model and then keeps track of both targets and distractors with an MOT philosophy. To the best of our knowledge, we are

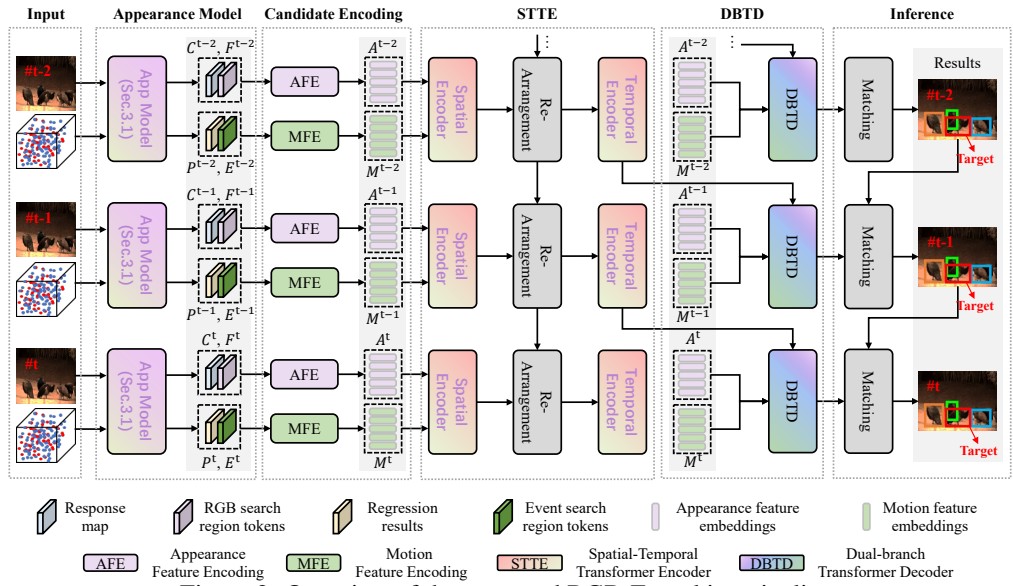

Figure 2: Overview of the proposed RGB-E tracking pipeline.

the first to introduce the MOT philosophy for the SOT task using RGB-E data. **(ii)** We propose three effective modules: a Candidate Encoding Module, a Spatial-Temporal Transformer Encoder and a Dual-branch Transformer Decoder. The appearance information as well as the motion cues within the RGB-E data can be fully exploited by the proposed modules for accurate candidate association. **(iii)** We show significantly improved state-of-the-art results of our proposed method on multiple RGB-E tracking benchmarks.

## 2 Related Work

**Visual object tracking:** The current prevalent tracking pipelines can be categorized into three groups: CNN-based trackers, CNN-Transformer trackers and Transformer-based trackers. CNN-based trackers utilize a Siamese network [17, 15] or Discriminative Correlation Filter (DCF) [8, 1] to address tracking tasks by matching templates and search regions. However, the inherent properties of CNNs limit their ability for global information exploration and interaction, thereby constraining the advancement of CNN-based trackers. In response, some CNN-Transformer trackers [33, 5, 28] employ attention mechanisms to establish global dependencies between template features and search features. But these hybrid CNN-Transformer trackers still independently extract features from templates and search regions using CNN networks, resulting in extracted features being unaware of the tracking target. To address this issue, several pure Transformer-based trackers [35, 32, 40] overcome the challenge by unifying feature extraction and feature relation modeling through a single Transformer backbone, leading to state-of-the-art tracking performance.

**RGB-E object tracking:** In recent years, there has been a growing interest among researchers in merging RGB frames and event streams for object tracking. Some researchers focus on exploring complementary information within RGB-E data via specially designed cross-modal interaction strategies [29, 24, 44]. For instance, Zhang et al. [29] proposed a cross-domain feature integrator to dynamically fuse visual cues from both the frame and event domains. Alternatively, CEUTtrack [24] proposed a one-stream framework based on the Transformer, which simultaneously addresses feature extraction, template-search relation modeling, and cross-modal interaction. Recently, some methods [13, 42, 31] aim to adapt the RGB tracking model to RGB-E tracking in the prompt learning manner. *However, existing RGB-E tracking frameworks cannot fully explore the abundant motion cues within the event stream, consequently limiting tracking performance in the presence of distractors.*

**Multi object tracking:** Multi-object tracking (MOT) aims to track multiple objects in a video sequence. Currently, the tracking-by-detection paradigm [7, 6], where an object detector is initially employed to locate all proposals, followed by an association network to match all of these objects, is gaining popularity for the MOT task. Additionally, some researchers have explored the joint-detection-and-tracking pipeline [23], aiming to achieve detection and tracking simultaneously in a

single stage. There are some approaches that aim to enhance the tracking performance of Single Object Tracking (SOT) via the use of an MOT philosophy. For instance, DMTrack [41] designed a lightweight detector and an explicit object association module to track both targets and distractors. KeepTrack [19] proposed a learnable candidate matching network and designed several mechanisms, including partial supervision, self-supervised learning and sample-mining, to address the problem of incomplete annotation in SOT training data. *However, these methods only use the RGB modality and overlook the importance of spatial-temporal relationships among candidates for matching candidates and tracklets.*

## 3    Method

As shown in Fig. 2, our framework first employs an appearance model to generate the potential proposals. Subsequently, several modules are proposed to identify targets and maintain tracking of all candidates to prevent tracking drift. Specifically, the appearance model predicts the target scores and bounding boxes for $M$ candidates of the $t - th$ current frame and $N$ candidates for each previous frame (see Sec.3.1). Secondly, a set of features are extracted for each candidate from $T$ previous frames, including target classification scores, appearance features, event embeddings and candidate locations. These features are then aggregated into appearance embeddings and motion embeddings for each proposal (see Sec.3.2). Thirdly, the STTE is employed to jointly model spatial-temporal relationships for each tracklet (see Sec.3.3). Fourthly, utilizing the DBTD, an $N \times M$ assignment matrix is calculated for matching tracklets from previous frames with candidates from the current frame (see Sec.3.4). It should be noted that not every tracklet has all the candidates in the previous $T$ frames due to occlusion, missing detection, etc. For illustrative purposes, we consider the situation that there are no missing candidates for each tracklet. Our method solves the cases with missing tracklets similarly to the typical multi-object tracking method [6] (see supplementary material Sec.B). In the following, we will describe the proposed tracking framework in detail.

### 3.1    Appearance Model

Here, we employ CEUTrack [24] as our appearance model. Specifically, the event streams are initially transformed into voxel representations through a voxelization operation [24]. Subsequently, given the initial locations and tracking results, we crop the template patch, the template voxel, the search patch and the search voxel, respectively. After that, the projection layers are adopted to transform the four inputs into token representations, which are then fed into the vanilla ViT [10] for joint feature extraction, cross-modal interaction and search-template matching. Finally, the tracking head, employing the same structure as that in OStrack [35], takes the concatenated RGB and event search region features from the backbone as input to predict the appearance tracking results. Please refer to the supplementary material (Sec.A.1) for additional details about the appearance model.

### 3.2    Candidate Embedding Module

The goal of CEM is to first select initial candidates similar to the target and filter out the most simple negative backgrounds, and then obtain the appearance embeddings as well as the motion embeddings. With the classification scores and regression offsets outputted by the appearance model, we generate $N$ candidates similar to the target for each previous frame by using Non-maximum Suppression (NMS) [20]. The appearance features of each candidate can be obtained by performing the PRoIAligh [22] on the RGB backbone features based on its corresponding location. We can represent the appearance features of each candidate in the $(t - T) - th$ frame as $F^{t-T} = \{f_N^{t-T} \in \mathbb{R}^d\}_{n=1}^N$ and represent their corresponding classification scores as $C^{t-T} = \{c_N^{t-T} \in \mathbb{R}^1\}_{n=1}^N$. Both the appearance features and the classification scores convey essential appearance cues for each candidate. To integrate these two types of information, we propose an Appearance Feature Encoding (AFE) layer. Specifically, AFE processes the backbone features $f_N^{t-T}$ via a single convolution layer to obtain more discriminative features $fe_N^{t-T} \in \mathbb{R}^d$ and employs several MLP layers on $c_N^{t-T}$ to generate the classification embeddings $ce_N^{t-T} \in \mathbb{R}^d$. These features are then combined as: $a_N^{t-T} = fe_N^{t-T} + ce_N^{t-T}$. The appearance feature embeddings for the $(t - T) - th$ frame are thus represented by $A^{t-T} = \{a_N^{t-T}\}_{n=1}^N$.

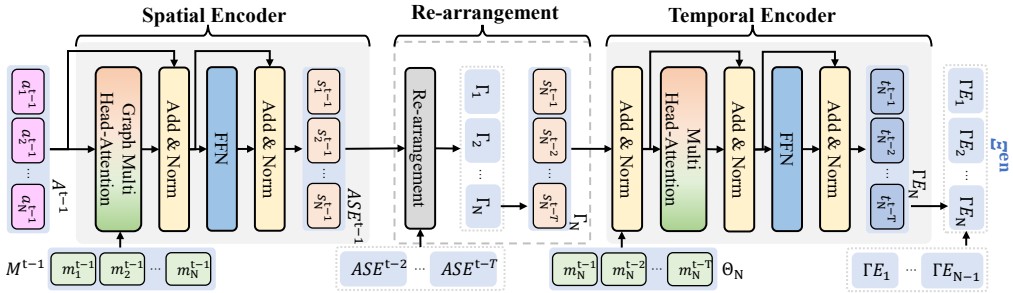

Figure 3: Architectures of the proposed Spatial-Temporal Transformer Encoder.

Additionally, we obtain the event features of each candidate in the $(t-T)-th$ frame as $E^{t-T} = \{e_N^{t-T} \in \mathbb{R}^d\}_{n=1}^N$ and represent their locations as $P^{t-T} = \{p_N^{t-T} \in \mathbb{R}^4\}_{n=1}^N$, where $p_N^{t-T} = \{x_N^{t-T}, y_N^{t-T}, w_N^{t-T}, h_N^{t-T}\}$ denotes the normalized bounding box coordinates. Both the event stream and the location set contain rich motion information about those candidates. To fuse these two types of features, a Motion Feature Encoding (MFE) layer, which has a similar structure of AFE, is first used to obtain the enhanced event representations $ee_N^{t-T} \in \mathbb{R}^d$ and location embeddings $pe_N^{t-T} \in \mathbb{R}^d$. Then these features are fused as: $m_N^{t-T} = ee_N^{t-T} + pe_N^{t-T}$. The motion feature embeddings of each candidate in the $(t-T)-th$ frame can be thus represented by $M^{t-T} = \{m_N^{t-T}\}_{n=1}^N$.

### 3.3 Spatial-Temporal Transformer Encoder

The proposed STTE aims at learning more discriminative feature representations for each tracklet and establishing effective relationships among objects in both spatial and temporal domains. The inputs for STTE include $T$ sets of appearance embeddings $\{A^{t-T}, ..., A^{t-1}\}$ and $T$ sets of motion representations $\{M^{t-T}, ..., M^{t-1}\}$. All of these embeddings will be first processed via a Spatial Encoder to construct the spatial relationships among candidates in each frame. Subsequently, these spatially encoded features are re-arranged to construct $N$ tracklets across $T$ frames, and their temporal relationships are established via the proposed Temporal Encoder.

**Spatial Transformer Encoder:** The proposed Spatial Encoder independently processes each frame to construct spatial correlations, and we take the $(t-1)-th$ frame as an example for illustration. It is difficult to establish distinctive spatial relationships by only using the appearance information. Differently, the motion information is more suitable to establish meaningful spatial affinities. Inspired by the graph attention networks [26], each candidate's appearance embedding is regarded as a node, and the edge weight between each nodes is defined by the motion information.

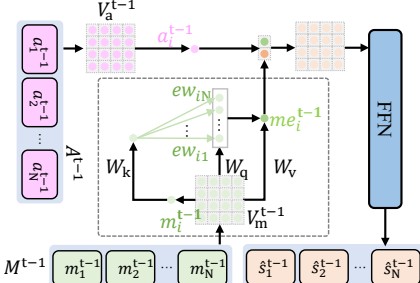

Figure 4: Architectures of proposed Graph Multi-head Attention Block.

Specially, at the $(t-1)-th$ frame, we consider each candidate's appearance and motion representations in $A^{t-1}$ and $M^{t-1}$ as two node sets $V_a^{t-1}$ and $V_m^{t-1}$, respectively. As shown in Fig. 3 (a), we use a complete bipartite graph $G_m^{t-1} = (V_m^{t-1}, E_m^{t-1})$ to model the object-level relations between these candidates. Here, $E_m^{t-1} = \{(u, v)|\forall u, v \in V_m^{t-1}\}$. Then, the edge weight between node $i$ and node $j$ in $V_m^{t-1}$ will be denoted as $e_{ij}$, which can be calculated through the inner product operation. More specifically, as that in a typical Transformer block, some normalization layers and linear transformations are first applied on these motion nodes, followed by an inner product calculation, to achieve $e_{ij}$. Formally,

$$e_{ij} = (W_k m_i^{t-1})^T (W_q m_j^{t-1}), \qquad (1)$$

where $W_k$ and $W_q$ are the linear transformations and $e_{ij}$ will be further normalized by the softmax fuction, obtaining

$$ew_{ij} = \frac{exp(e_{ij})}{\sum_{k \in V_m^{t-1}} exp(e_{ik})}. \qquad (2)$$

With the edge weights passed from all nodes in $V_{\mathrm{m}}^{\mathrm{t}-1}$ to the $i-$th node in $V_{\mathrm{m}}^{\mathrm{t}-1}$, the aggregated representation of the $i-$th node in $V_m^{\mathrm{t}-1}$ can be transformed by:

$$me_i^{\mathrm{t}-1} = \sum_{j \in V_{\mathrm{m}}^{\mathrm{t}-1}} ew_{ij} W_{\mathrm{v}} m_j^{\mathrm{t}-1}. \tag{3}$$

where $W_{\mathrm{v}}$ is a matrix for linear transformation. It should be noted that we adopt the multi-head attention structure to improve the discriminability of graph attention learning.

Finally, we fuse the aggregated features $me_i^{\mathrm{t}-1}$ with the appearance features $a_i^{\mathrm{t}-1}$ to obtain a more powerful feature representation:

$$\hat{s}_i^{\mathrm{t}-1} = FFN(cat(me_i^{\mathrm{t}-1}, a_i^{\mathrm{t}-1})), \tag{4}$$

where $cat(\cdot)$ represents vector concatenation, $FFN(\cdot)$ denotes the feedforward neural network. The final spatial encoded features $ASE^{\mathrm{t}-1} = \{s_1^{\mathrm{t}-1}, ..., s_N^{\mathrm{t}-1}\}$ are obtained by further employing the residual connections and two FFN layers as that in a typical Transformer block. These spatial encoded features are re-arranged to N tracklet sets $\{\Gamma_1, ..., \Gamma_N\}$ of all candidates through the T frames, where $\Gamma_N = \{s_N^{\mathrm{t}-\mathrm{T}}, ..., s_N^{\mathrm{t}-1}\}$. Meanwhile, the motion embeddings are also re-arranged to N tracklet sets $\{\Theta_1, ..., \Theta_N\}$, where $\Theta_N = \{m_N^{\mathrm{t}-\mathrm{T}}, ..., m_N^{\mathrm{t}-1}\}$.

**Temporal Transformer Encoder:** These N tracklet sets are further encoded by a Temporal Transformer Encoder. Here, we take $N - \mathrm{th}$ tracklet as an example for illustration. As shown in Fig. 3, we first fuse the spatial encoded features $\Gamma_N$ with the motion feature set $\Theta_N$ by the element-wise addition operation, and then employ the multi-head attention to calculate the attention weights $A_N \in \mathbb{R}^{\mathrm{T} \times \mathrm{T}}$, thus generating the attention-weighted features. After that, these weighted features are processed through two FFN layers and residual connections to obtain the final output $\Gamma E_N$, which consists of the $N - \mathrm{th}$ tracklet's feature representations $\{t_N^{\mathrm{t}-\mathrm{T}}, ..., t_N^{\mathrm{t}-1}\}$. The outputs of the Temporal Transformer Encoder $\{\Gamma E_1, ..., \Gamma E_N\}$ are re-arranged to $\Xi^{\mathrm{en}} \in \mathbb{R}^{\mathrm{NT} \times \mathrm{d}}$.

### 3.4   Dual-branch Transformer Decoder

The proposed DBTD generates the assignment matrix $A$ by using the output of the Transformer encoder $\Xi^{\mathrm{en}}$ and the features of M candidates in the current frame. Initially, we generate the spatial-encoded feature set $\{s_1^{\mathrm{t}}, ..., s_M^{\mathrm{t}}\}$ and re-arrange it to $\Xi_{\mathrm{en}}^{\mathrm{can}} \in \mathbb{R}^{\mathrm{M} \times \mathrm{d}}$ as that in the Sec.3.3. Then, $\Xi_{\mathrm{en}}^{\mathrm{can}}$ is duplicated N times, resulting in $\Xi_{\mathrm{en}}^{\mathrm{can}} \to \hat{\Xi}_{\mathrm{en}}^{\mathrm{can}} \in \mathbb{R}^{\mathrm{NM} \times \mathrm{d}}$.

After that, the proposed Spatail-temporal Decoder follows the standard Transformer framework, which takes the spatial-encoded features of the current frame $\hat{\Xi}_{\mathrm{en}}^{\mathrm{can}}$ as the query, and uses the encoded features of the previous frame $\Xi^{\mathrm{en}}$ as the key and

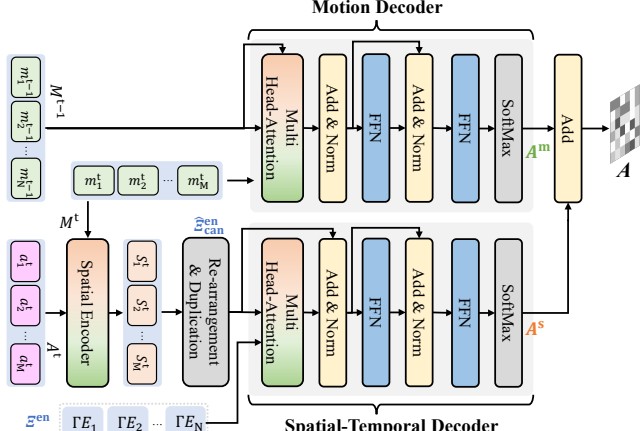

Figure 5: Architectures of the proposed Dual-branch Transformer Decoder.

value. The Multi-Head Attention mechanism [24] is calculated for $\hat{\Xi}_{\mathrm{en}}^{\mathrm{can}}$ and $\Xi^{\mathrm{en}}$ to generate attention weights. The output passes through two FFN layers and residual connections, generating the output tensor $\mathbb{R}^{\mathrm{NM} \times \mathrm{d}}$. The output of the appearance decoder can be processed through an FFN and a softmax layer to generate the assignment matrix $A^{\mathrm{s}} \in \mathbb{R}^{\mathrm{N} \times \mathrm{M}}$.

Moreover, to better match each tracklet with the candidates in the current frame, the motion information of the $(\mathrm{t} - 1) - \mathrm{th}$ frame is utilized as the motion information for the tracklet. The Motion Decoder employs the same structure as that in the appearance decoder to obtain the assignment matrix $A^{\mathrm{m}} \in \mathbb{R}^{\mathrm{N} \times \mathrm{M}}$. The final assignment matrix $A$ can be obtained by $A = A^{\mathrm{s}} + A^{\mathrm{m}}$.

## 3.5 Object Association

During inference, we first employ the appearance model to generate candidates of each frame. If only one target with a high score is present in both the previous and current frames, this candidate is selected as the target, and the candidate matching model is omitted to reduce computational costs and accelerate inference. In contrast, when multiple candidates exist during tracking, the assignment matrix $A$ is calculated using the proposed candidate matching model. After that, a threshold $\tau_{\text{th}}$ is adopted on $A$ to remove the ambiguous correspondence. Finally, we match the predicted boxes and the candidate boxes using the Hungarian algorithm [14]. The candidates that do not match any tracklet will be assigned a new ID, and the tracklets that do not match any of the detections in the past consecutive T frames will be terminated. T is experimentally set to 15. Please refer to the supplementary material (Sec.B.3) for more details.

# 4 Experiment

Table 1: Comparison with state-of-the-art trackers on COESOT [24], FE108 [38] and VisEvent [29]. The numbers with red and blue colors indicate the best and the second best results, respectively.

| Method | Source | Backbone | Type | FE108 | | VisEvent | | | COESOT | | |
|---|---|---|---|---|---|---|---|---|---|---|---|
| | | | | RSR | RPR | PR | NPR | SR | PR | NPR | SR |
| STNet[36] | CVPR22 | - | Event | 58.5 | 89.6 | 49.2 | - | 35.2 | - | - | - |
| MonTrack[43] | NeurIPS22 | - | Event | 63.3 | 90.7 | - | - | - | - | - | - |
| DANet[11] | TIP23 | Res18 | Event | 56.7 | 89.2 | 54.5 | - | 39.8 | - | - | - |
| HDETrack[30] | CVPR24 | ViT-B | Event | 59.8 | 92.2 | - | - | - | 59.0 | 59.0 | 52.3 |
| DiMP*[1] | ICCV19 | Res50 | RGB-E | 57.1 | 85.1 | 67.0 | 58.1 | 47.8 | 67.1 | 65.9 | 58.9 |
| PrDiMP*[9] | CVPR20 | Res50 | RGB-E | 55.2 | 86.8 | 65.3 | 57.7 | 47.6 | 65.0 | 64.0 | 57.9 |
| SiamRCNN*[27] | CVPR20 | Res101 | RGB-E | - | - | 68.0 | 62.6 | 52.7 | 67.5 | 66.3 | 60.9 |
| TrDiMP*[28] | CVPR21 | Res50 | RGB-E | 60.3 | 91.2 | - | - | - | 66.9 | 65.8 | 60.1 |
| TransT*[5] | CVPR21 | Res50 | RGB-E | 63.9 | 93.0 | - | - | - | 67.9 | 66.6 | 60.5 |
| ToMP*[18] | CVPR22 | Res101 | RGB-E | 61.8 | 91.1 | - | - | - | 67.2 | 66.0 | 59.9 |
| FENet[38] | ICCV21 | Res18 | RGB-E | 63.1 | 91.8 | - | - | - | - | - | - |
| CEUTrack[24] | ArXiv22 | ViT-B | RGB-E | 55.6 | 84.5 | 71.8 | 66.4 | 53.5 | 70.5 | 69.0 | 62.0 |
| HRCEUTrack-B[45] | ICCV23 | ViT-B | RGB-E | - | - | - | - | - | 71.9 | 70.2 | 63.2 |
| HRCEUTrack-L[45] | ICCV23 | ViT-L | RGB-E | - | - | - | - | - | 73.8 | 71.9 | 65.0 |
| HRMonTrack-T[45] | ICCV23 | ViT-B | RGB-E | 66.3 | 95.3 | - | - | - | - | - | - |
| HRMonTrack-B[45] | ICCV23 | ViT-L | RGB-E | 68.5 | 96.2 | - | - | - | - | - | - |
| AFNet[37] | CVPR23 | Res18 | RGB-E | - | - | - | - | - | 67.8 | - | 59.2 |
| ViPT†[42] | CVPR23 | ViT-B | RGB-E | - | - | 76.6 | 73.0 | 60.8 | 73.9 | 72.2 | 65.7 |
| SDSTrack[31] | CVPR24 | ViT-B | RGB-E | - | - | 79.3 | 75.5 | 62.6 | - | - | - |
| OneTrack†[13] | CVPR24 | ViT-B | RGB-E | - | - | 78.1 | 75.6 | 63.2 | - | - | - |
| SeqTrackv2-B256†[4] | ArXiv24 | ViT-B | RGB-E | - | - | 79.9 | 76.5 | 63.7 | - | - | - |
| SeqTrackv2-L256†[4] | ArXiv24 | ViT-L | RGB-E | - | - | 80.6 | 77.8 | 65.2 | - | - | - |
| CSAM-T† | 2024 | ViT-T | RGB-E | 66.7 | 95.5 | 76.1 | 72.4 | 61.5 | 73.3 | 70.5 | 63.6 |
| CSAM-B† | 2024 | ViT-B | RGB-E | 70.5 | 97.1 | 81.6 | 78.6 | 65.9 | 76.7 | 74.8 | 68.1 |

## 4.1 Implementation details

Our proposed CSAM is implemented in Python 3.8 using PyTorch 1.7.1. The CSAM training is conducted on two Nvidia RTX 3090 GPUs. For inference, we test our tracker on a single Nvidia RTX 3090 GPU. The search region is $4^2$ times the target object area and resized to a resolution of 256×256 pixels, whilst the template is $2^2$ times the target object area and resized to $128 \times 128$ pixels.

**Architectures:** We instantiate two models of CSAM: CSAM-T and CSAM-B, by varying the backbone network in the appearance model, i.e., ViT-Tiny and ViT-Base. We initialize ViT-Tiny using the weights from DeiT-tiny[25], and the backbone weights ViT-B are initialized with corresponding MAE encoders[12]. In the candidate matching network, both the proposed STTE and DBTD apply one individual layer. Please refer to the supplementary material (Sec.A.1 and Sec.B.1) for more details.

**Training:** The training of our CSAM comprises three parts. In the first part, the following three loss functions are adopted: the focal loss for classification, and L1 loss and GIOU loss for bounding box regression[35]. We employ the same training setting as that in OSTrack[35] to train an RGB tracker. Secondly, we employ the same training setting as that in HRCEUTrack[45] to train the

appearance model. In the third part, the parameters of the appearance model are fixed and other parameters in our proposed framework are set to be trainable. Since only the targets' locations are provided in the existing RGB-E tracking dataset, the partial supervision loss and self-supervised loss in KeepTrack [19] are employed to supervise the assignment matrix $A$ generated by our proposed model. Please refer to the supplementary material (Sec.A.2 and Sec.B.2) for additional details about the implementation details.

## 4.2 Evaluation datasets and metrics

**Dataset:** We evaluate the performance of our proposed CSAM on three large-scale RGB-E single object tracking datasets: VisEvent [29] FE108 [38] and COESOT [24]. These three datasets were captured using DAVIS346, with a spatial resolution of $346 \times 260$, a dynamic range of 120 dB and the minimum latency of 20 $\mu$s. The COESOT [24] dataset comprises 578K RGB-E pairs, divided into 827 and 527 sequences for training and testing, respectively. These sequences were collected from both indoor and outdoor scenarios, covering a range of 90 classes and 17 challenging attributes. The FE108 [38] dataset contains 108 RGB-E sequences, which capture 21 different types of objects. It is divided into 76 and 32 sequences for training and testing, respectively. VisEvent [29] dataset collects 820 RGB-E video pairs, divided into 500 and 320 sequences for training and testing, respectively. Following [37], after removing sequences that miss event data or have misaligned timestamps, the VisEvent dataset includes 377 sequences for training and 172 for testing.

**Metrics:** In FE108 [38], we use representative success rate (RSR) and representative precision rate (RPR) to evaluate all trackers. In COESOT [24] and VisEvent [29], we use success rate (SR), precision rate (PR) and normalized precision rate (NPR) for evaluation.

## 4.3 Comparisons with State-of-the-art Methods

To show the effectiveness of the proposed method, we evaluate and compare our CSAM with several state-of-the-art trackers, including 4 Event trackers and 17 RGB-E trackers. As shown in Table. 1. * denotes that the RGB trackers are extended to RGB-E trackers via the early fusion approach. † and * denotes that the model is pre-trained on RGB tracking datasets.

**Results on FE108:** As shown in Table. 1, our proposed CSAM-B outperforms other top-performing trackers, such as HRMonTrack-B [44], HDETrack [30] and TransT [5], with a clear margin, and achieves the best performance with an RSR score of 70.5%. and an RPR score of 97.1%. Even when compared to HRMonTrack-B, which has already obtained impressive tracking performance, our approach demonstrates notable improvements, with a 2.0% increase in RSR and a 0.9% increase in RPR. These comparisons fully demonstrate the effectiveness of tracking multiple candidates for robust tracking.

**Results on VisEvent:** From Table. 1, we find that our method sets a new state-of-the-art score on VisEvent. First, our proposed framework outperforms the Event trackers, e.g., STNet [36] and DANet [11], by a clear margin. Secondly, compared with appearance trackers SDSTrack [31] and ViPT [42], our model further improves the PR score by 4.1% and 1.6% in NPR scores, respectively. This enhancement is attributed to our model's comprehensive utilization of both appearance and motion information, enabling effective tracking of targets and distractors. Thirdly, our tracker surpasses the previous best tracker SeqTrackv2-L [4], which demonstrates that our method has a stronger capability in handling various challenges.

**Results on COESOT:** As shown in Table. 1, our proposed CSAM-B achieves a PR score of 76.7% and a SR score of 68.1%, surpassing recent state-of-the-art trackers. Compared with the most competitive RGB-E tracker ViPT [42], our CSAM-B achieves performance gains of 2.4% in NPR score. These results meet our expectation that the exploration of spatial-temporal relationships from the appearance cues as well as the motion cues can effectively match the candidates and tracklets for SOT task.

**Speed Analysis:** As shown in Table 2, despite increased computational costs and parameters for simultaneous target and distractor tracking, CSAM maintains real-time performance on the RTX 3090 GPU. It strikes a good balance between resource consumption and efficacy compared to competitors. Compared with the appearance tracker, our CSAM introduces limited computation costs, while significantly improving the tracking performance. Consequently, CSAM-B achieves an

average running speed of 53 frames per second (FPS). We also notice CSAM's superior performance compared with the second best RGB-E tracker SeqTrackv2-L256.

Table 2: CSAM-B's efficiency analysis on VisEvent with a fixed candidate count of 4 for FLOPS calculation.

|  | ViPT | SeqTrackv2-B256 | SeqTrackv2-L256 | Appearance Tracker | CSAM-B |
|---|---|---|---|---|---|
| PR/SR | 76.6/60.8 | 79.9/63.7 | 80.6/65.2 | 75.3/60.6 | 81.6/65.9 |
| FPS | 75 | 40 | 15 | 75 | 53 |
| Model size (M) | 93.3 | 89 | 309 | 92.5 | 106.9 |
| FLOPS (G) | 52.1 | 66 | 232 | 62.7 | 83.2 |

## 4.4  Ablation Study

To verify the effectiveness of our designed framework, we perform ablation analysis to evaluate different components in our method by using the COESOT test set [24]. Please refer to the supplementary material (Sec.C) for more ablation experiments.

Table 3: Experiment results of different variants for Candidate Encoding Module (CEM), Spatial-Temporal Transformer Encoder (STTE) and Dual-Branch Transformer Decoder (DBTD). Here, 'AppModel', 'SE', 'TE', 'STD' and 'MD' denote the appearance model, Spatial encoder, Temporal encoder, Spatial-temoral decoder and Motion decoder, respectively.

|  | AppModel | AFE | MFE | OTE | SuperGlue | SE | TE | OTD | STD | MD | SR | PR |
|---|---|---|---|---|---|---|---|---|---|---|---|---|
|  | ✓ |  |  |  |  |  |  |  |  |  | 65.5 | 74.8 |
| CEM | ✓ |  |  |  |  | ✓ | ✓ |  | ✓ | ✓ | 67.1 | 75.9 |
|  | ✓ | ✓ |  |  |  | ✓ | ✓ |  | ✓ | ✓ | 67.3 | 76.4 |
|  | ✓ |  | ✓ |  |  | ✓ | ✓ |  | ✓ | ✓ | 67.5 | 76.3 |
| STTE | ✓ | ✓ | ✓ |  |  |  |  |  | ✓ | ✓ | 66.2 | 75.3 |
|  | ✓ | ✓ | ✓ | ✓ |  |  | ✓ |  | ✓ | ✓ | 66.8 | 75.8 |
|  | ✓ | ✓ | ✓ |  | ✓ |  | ✓ |  | ✓ | ✓ | 67.1 | 75.9 |
|  | ✓ | ✓ | ✓ |  |  | ✓ |  |  | ✓ | ✓ | 67.4 | 76.2 |
|  | ✓ | ✓ | ✓ |  |  |  | ✓ |  | ✓ | ✓ | 66.6 | 75.5 |
| DBTD | ✓ | ✓ | ✓ |  |  | ✓ | ✓ | ✓ |  |  | 66.7 | 75.3 |
|  | ✓ | ✓ | ✓ |  |  | ✓ | ✓ |  | ✓ |  | 67.7 | 76.3 |
|  | ✓ | ✓ | ✓ |  |  | ✓ | ✓ |  |  | ✓ | 66.4 | 75.5 |
| CSAM | ✓ | ✓ | ✓ |  |  | ✓ | ✓ |  | ✓ | ✓ | 68.1 | 76.7 |

**Effectiveness of the proposed CEM:** To investigate the impact of our proposed CEM, several versions of our proposed method are provided, including ①: Removing the AFE and MFE sub-modules in CEM block. ②: Removing the AFE sub-module in CEM block. ③: Removing the MFE sub-module in CEM block. As can be seen in Table 3, the tracking performance degrades after removing AFE or MFE sub-modules, which demonstrates the necessity of embedding appearance and motion information from the classification scores and bounding box coordinates.

**Effectiveness of the proposed STTE:** To further verify the effectiveness of the proposed STTE, several variants are proposed, including ①: Removing the spatial encoder (SE) and temporal encoder (TE) . ②: Replacing the spatial encoder by the original Transformer Encoder (OTE) block[10]. ③: Replacing the spatial encoder by SuperGlue[19]. ④: Removing the temporal encoder in STTE block. ⑤: Removing the spatial encoder in STTE block. As can be seen in Table 3, the tracking performance experiences a significant decline upon the removal of temporal encoder or spatial encoder, which confirms the necessity of spatial-temporal relationships in enhancing feature representations of the candidates. Furthermore, compared with several existing methods, e.g., OTE and SuperGlue, the proposed spatial encoder takes full advantage of the appearance information as well as motion information in constructing robust spatial correlations, thus achieving better results.

**Effectiveness of the proposed DBTD:** To further verify the effectiveness of the proposed DBTD, several variants are proposed, including ①: Replacing the spatial-temporal decoder (STD) by the original Transformer Decoder (OTD) block[10]. ②: Removing the motion decoder (MD) in DBTD block. ③: Removing the spatial-temporal decoder in DBTD block. As can be seen in Table 3, not using spatial-temporal decoder or motion decoder substantially deteriorates the performance. This demonstrates that the exploration of both appearance and motion information is crucial for ensuring robust tracking.

**Visualizations of the MOT philosophy** Based on the proposed candidate matching network, CSAM-B can track both the targets and distractors. As shown in Fig. 6, (•, •, •) denote the target and the candidates in each frame. An object disappears from the scene if none of the current candidates are associated with it. The MOT philosophy in CSAM-B can effectively suppress the negative influence of distractors for tracking.

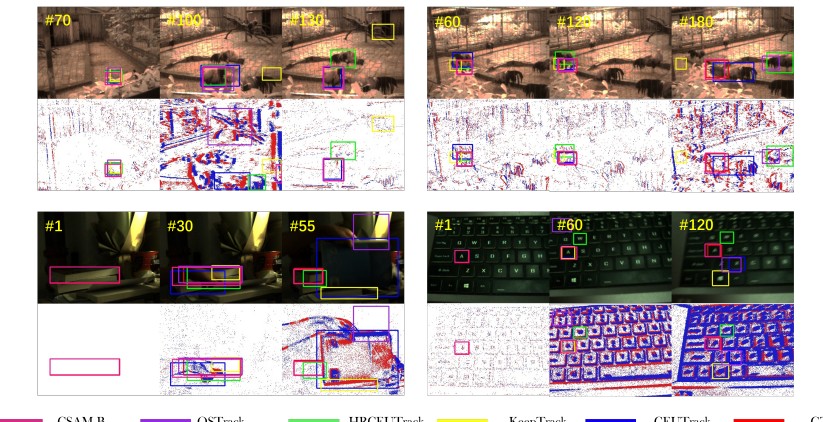

Figure 6: Visualizations of the MOT philosophy. (•, •, •) denote the target and the candidates in each frame. Red bounding boxes refer to the tracking results. Event images are used for visualizations only.

## 5 Conclusion

In this paper, a novel RGB-E tracking framework with MOT philosophy has been proposed in order to keep track of both targets and distractors to robustly track a single object. Specifically, a Spatial-Temporal Transformer Encoder is proposed to establish a rich temporal-spatial relationships by using appearance information in combination with motion information. Furthermore, by formulating the tracklets and candidates with the appearance features and motion embeddings, the affinities of the tracklets and candidates are explicitly modeled and leveraged. We conduct comprehensive experimental validation and analysis of our approach on three RGB-E object tracking benchmarks and produce new state-of-the-art results.

**Limitation:** The current method is dedicated to constructing an effective framework for RGB-E tracking with MOT philosophy, but it pays less effort to improve the efficiency of the supervision signals, which we will consider in future work.

## 6 Acknowledgments

This work was supported in part by the State Key Laboratory of Reliability and Intelligence of Electrical Equipment under Grant EERI KF2022005, in part by the Hebei University of Technology, in part by the National Natural Science Foundation of China under Grant 61803290 and Grant 61773301, in part by China Postdoctoral Science Foundation under Grant 2023M742745, and in part by the Natural Science Foundation of Shaanxi Province under Grant 2019JQ-312.

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

# Appendix

In this supplementary material, we first provide details of the appearance model in Sec. A. Subsequently, we introduce the training and inference details of the candidate matching network in Sec. B. More experimental results are shown in Sec. C.

## A  Appearance Model

In this section, we first describe the architectural details of the appearance model, which consists of the input representation, the projection layer, the backbone network and the tracking head. Then, we introduce the training details of the Appearance model.

### A.1  Architectural details of the appearance model

In this paper, we employ an existing RGB-E tracker, i.e., CEUTrack[24], as our appearance model, which can be divided into four main parts: input representations, projection layer, backbone network and head network, as shown in Fig. 7.

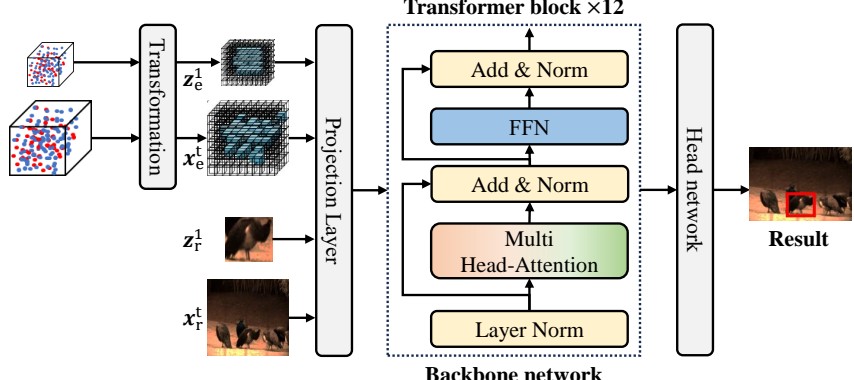

Figure 7: Architectures of the appearance model.

**Input Representation:** We employ the method used in CEUtrack [24] to process the input RGB images and event data. Specifically, the RGB and event data can be represented as follows:

$$\begin{aligned} \mathcal{I}_t &= \{x_i, y_j\}, \\ \mathcal{E}_t &= \{e_s\} = \{[x_s, y_s, t_s, p_s]\}, \end{aligned} \tag{5}$$

where $\mathcal{I}_t$ denotes the t-th RGB image, $(x_i, y_j)$ represents the spatial coordinates with $i \in \{1, \ldots, W\}, j \in \{1, \ldots, H\}$. Here, $W, H$ are the width and height of the RGB images. $\mathcal{E}_t$ is composed of multiple event points $e_s$, where $s \in \{1, \ldots, S\}$. S represents the total number of event points. $(x_s, y_s)$ are spatial coordinates of the event point, $t_s$ is the timestamp of each event point, and $p_s$ represents the polarity of the corresponding event point, i.e., an ON or OFF event.

For the RGB frames, we first crop the template and search region, each containing 2 times and 4 times more extensive regions than the provided annotations, respectively. Then, the template and search patches of the RGB modality are resized into $z_1^r \in \mathbb{R}^{H_z \times W_z \times 3}$ and $x_t^r \in \mathbb{R}^{H_x \times W_x \times 3}$, respectively. In this paper, the template patch is uniformly resized into $128 \times 128 \times 3$, while the search patch is resized to $256 \times 256 \times 3$.

For the event data, to transform the event points into voxels, we first split the whole event stream $\mathcal{E}$ into $\mathcal{E}_t$ based on the time interval of the RGB frames, obtaining $\{\mathcal{E}_1, ..., \mathcal{E}_t\}$. T is the total number of RGB frames. Then, we transform each event set $\mathcal{E}_t$ to volumetric representations $\mathcal{V}_t = \{[x_o, y_p, t_q, f_{opq}]\}_{l=1}^Q$ via the voxelization process in SECOND [34], where Q is the total voxel number of the current voxel set. $(x_o, y_p, t_q) \in \mathbb{R}^{Q \times 3}$ and $f_{opq} \in \mathbb{R}^{Q \times 16}$ denote the 3D coordinates and corresponding feature representations, respectively. Subsequently, the voxel grids of the event modality, whose coordinates are not in the template or search region, are removed. We further select the top $4096$ and $1024$ voxel grids in the cropped template and search voxels based on the density of

event points, respectively, thereby obtaining the template voxels $z_1^e \in \mathbb{R}^{1024 \times 16}$ and search voxels $x_t^e \in \mathbb{R}^{4096 \times 16}$.

**Projection layer:** We employ a projection layer, consisting of four parallel $Conv\_BN\_Relu$ blocks, to transform the input RGB and event data into token sequences. Specifically, for the RGB frame, similar to ViT[10], the template $z_1^r$ and search patches $x_t^r$ are projected to feature embeddings $F_z^r \in \mathbb{R}^{N_z \times C}$ and $F_x^r \in \mathbb{R}^{N_x \times C}$ by using two non-shared $16 \times 16$ convolution layers, where $N_z = \frac{W_z}{16} \frac{H_z}{16}$ and $N_x = \frac{W_x}{16} \frac{H_x}{16}$. For the event modality, the search voxel $x_t^e$ and template voxel $z_1^e$ are projected to feature embeddings $F_x^e \in \mathbb{R}^{N_x \times C}$ and $F_z^e \in \mathbb{R}^{N_z \times C}$ by using two non-shared $4 \times 4$ convolution layers. Furthermore, the position embeddings in ViT are further added to those feature embeddings.

**Backbone network:** The template and search tokens of the RGB modality and the event modality will be first concatenated as $U_f$ and then fed into the vanilla ViT, which consists of 12 stacked Transformer layers, to extract RGB features. Each Transformer layer consists of two layernorms (LN), the multi-head self-attention (MHSA) block, multi-layer perceptron (MLP), and some residual connections. The detailed computation process can be formulated as:

$$
\begin{aligned}
\text{MHSA}(Q, K, V) &= \text{Softmax}\left(\frac{\mathbf{Q}\mathbf{K}^\top}{\sqrt{d_k}}\right) \cdot \mathbf{V}, \\
\tilde{U}_f &= U_f + \text{MHSA}\left(LN_1\left(U_f, U_f, U_f\right)\right), \\
\overline{U}_f &= \tilde{U}_f + \text{MLP}\left(LN_2\left(\tilde{U}_f\right)\right).
\end{aligned}
\tag{6}
$$

In our experiments, we employ ViT-Base[10] and ViT-Tiny[10] as our backbone network, resepctively.

**Tracking Head:** To obtain the locations of the candidates, we employ a standard tracking head in OSTrack [35] to directly estimate the center positions and scales of the bounding boxes. Specifically, the tracking head contains three branches that predict the classification score, center offset, and the size of bounding boxes, respectively. Each branch consists of four $3 \times 3$ $Conv\_BN\_RELU$ and $1 \times 1$ $Conv$ layer, respectively.

### A.2 Training details of the appearance model

To train our proposed appearance model, we employ three loss functions: focal loss [16] for classification, L1 loss and GIOU loss [21] for bounding box regression. The overall loss function can be written as:

$$
L = \lambda_1 L_{\text{focal}} + \lambda_2 L_{L1} + \lambda_3 L_{\text{giou}},
\tag{7}
$$

where the hyper-parameters $\lambda_1$, $\lambda_2$, and $\lambda_3$ are set as 1, 1, and 14, respectively.

In the training phase, as illustrated in Sec. 4.1, we first employ the same training setting as that in OSTrack[35] to train an RGB tracker. Secondly, we use the RGB-E tracking datasets to train our appearance model. For COESOT [24], we train our proposed appearance model by using the training subset of COESOT. For VisEvent [29] and FE108 [38], we train our proposed appearance model by using the training subset of VisEvent. Horizontal flip and brightness jittering are used for data augmentation during model training. Our appearance model is optimized by the AdamW optimizer with a weight decay of $1 \times 10^{-4}$ for 50 epochs. The initial learning rate for the backbone and other parameters were set to $4 \times 10^{-5}$ and $4 \times 10^{-4}$, respectively.

## B  Candidate matching model

In this section, we provide detailed information of the architectural details as well as the training and inference details of the candidate matching model.

### B.1  Architectural details of the candidate matching network

**Candidate Embedding Module: Adjust layer:** Based on the classification and regression results of the appearance model, we can extract the features of the candidates within each frame. For CSAM-B and CSAM-T, a projection layer is employed to adjust the dimensions of the RGB and event features from the backbone network to 192 and 256, respectively.

**STTE and DBTD:** In the candidate matching model, both the proposed STTE and DBTD apply one individual layer. For CSAM-B, each attention block in the proposed STTE and DBTD employs multi-head attention layers with 6 heads. For CSAM-T, each attention block in the proposed STTE and DBTD employs multi-head attention layers with 3 heads.

### B.2   Training details of the candidate matching model

In the existing RGB-E datasets, only the target objects and the corresponding locations are provided for training. To effectively supervise the learning of the candidate matching model, the partial supervision loss and the self-supervision loss in KeepTrack [19] are employed.

**Partially Supervised Loss:** Here, we formulate the problem of target candidate association across two subsequent frames as, obtaining the affinity matrix $A$ between the two candidate sets. If the target candidate $v_i^t$ corresponds to $v_j^{t-1}$, $A_{ij} = 1$, otherwise $A_{ij} = 0$. For each consecutive frame in a video sequence, we retrieve the single candidate corresponding to the annotated target. For the candidates $\{v_i^{t-T}, ..., v_i^t\}$, the assignment matrix $A$ can be obtained by the proposed candidate matching network, which reflects the association between $v_i^t$ and $v_i^{t-1}$. The supervised loss is then given by the negative log-likelihood of the assignment probability,

$$L_{\text{sup}} = -\log \boldsymbol{A}_{i,i}. \tag{8}$$

**Self-Supervised Loss:** To improve the robustness of the candidate matching network, the appearance model is first employed to predict the candidate sets $\mathcal{V}^t$ and its corresponding ground-truth association set $\mathcal{C} = \{(i, i)\}_{i=1}^N$ from any given frame. Then, a series of candidate sets $\{\mathcal{V}^{t-T}, ..., \mathcal{V}^{t-1}\}$ are generated from $\mathcal{V}^t$ by feature augmentation. The feature augmentation involves randomly translating the locations of the candidates, randomly adjusting the classification scores of the candidates, and transforming the given image before extracting the multi-modal features. The self-supervised loss is given by,

$$L_{\text{self}} = \sum_{(i,j) \in \mathcal{C}} -\log \boldsymbol{A}_{i,j}. \tag{9}$$

Finally, we combine both self-supervised loss $L_{\text{self}}$ and partially supervised Loss $L_{\text{sup}}$ as $L_{\text{total}} = L_{\text{self}} + L_{\text{sup}}$.

To train the candidate matching network, we use the same training data for the appearance model. The appearance model is first employed to generate the score maps, the search regions and the regression results of each sequence, thereby locating the candidates. During the training of the candidate matching network, the weights of the appearance model will be frozen. Our candidate matching network is optimized by the Adam optimizer with a weight decay of 0.2 for 15 epochs. The initial learning rate is set to $4 \times 10^{-5}$.

### B.3   Inference details of the candidate matching netwrok

In this sub-section, we provide the detailed algorithm that describes the candidate matching association. During the inference, we first generate several candidates via the appearance model. Then, based on the affinity matrix $A$ produced by the candidate matching network, we check whether a target candidate matches any of those previously detected objects. Only the tracklet-candidate pairs with affinity larger than $\tau_{th} = 0.75$ can be associated. After that, a straightforward Hungarian algorithm [14] is utilized to generate the tracking output. If unmatched candidates exist, they will be connected to newly initialized trajectories. Finally, we check whether the object previously selected as the target is still visible in the current frame. If the previous tracklet of the target is visible in the current frame, we select the candidate which is matched to the previous tracklet of the target as the new target in the current frame. If the previous tracklet of the target is invisible in the current frame, we check the classification scores of other candidates. When a candidate with the highest classification score is greater than the threshold $\zeta = 0.25$, we select this candidate as the new target. When the classification scores of all candidates are lower than the threshold $\zeta = 0.25$, we determine that the target is not visible in the current frame and search for the target again in the next frame.

## C   Experiments

We provide more details to complement the state-of-the-art comparisons presented in the paper.

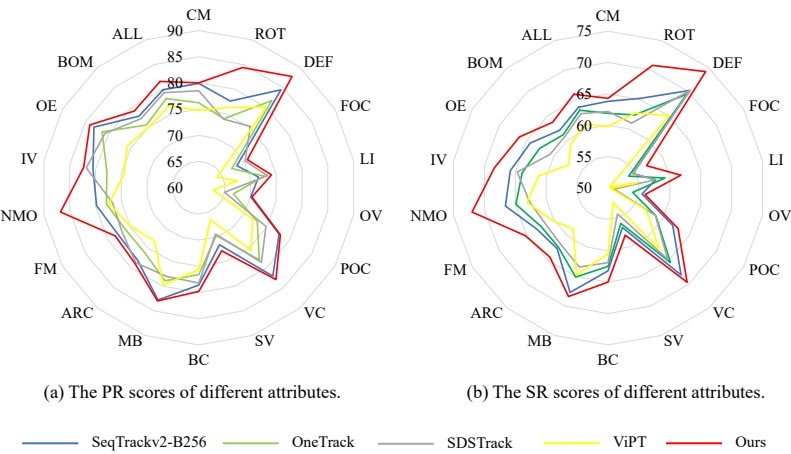

(a) The PR scores of different attributes.  (b) The SR scores of different attributes.

———— SeqTrackv2-B256  ———— OneTrack  ———— SDSTrack  ———— ViPT  ———— Ours

Figure 8: The SR and PR scores of the proposed CSAM and other RGB-E trackers under different attributes on VisEvent.

## C.1 Attributes Performance

**Details of VisEvent:** To further verify the effectiveness of our proposed CSAM, we further analyze the attribute-based performance on VisEvent [29]. VisEvent annotates each testing video sequence with 17 attributes to facilitate researchers in assessing the performance of their trackers under specific challenging scenes. These challenges encompass full occlusion (FOC), deformation (DEF), rotation (ROT), fast motion (FM), partial occlusion (POC), low illumination (LI), scale variation (SV), background object motion (BOM), motion blur (MB), overexposure (OE), camera motion (CM), out of the view (OV), Viewpoint change (VC), Background clutter (BC), illumination variation (IV), no motion (NMO) and aspect ratio change (ARC). As shown in Fig. 8, we analyze the attribute-based performance on VisEvent [29]. For clarity, we only illustrate 4 RGB-E trackers, including ViPT [42], SDSTrack [31] OneTrack [13] and SeqTrackv2-B256 [4]. All of these compared trackers employ the same training subset and backbone network. From the results, we can see that our proposed method performs the best in all annotated attributions.

Compared with existing RGB-E trackers, our approach exhibits significant improvements, particularly in cases LI, CM, NMO and OE. For the above-mentioned challenges, the data of a certain modality cannot contain valid target information for tracking. Our proposed method still performs well, which demonstrates the ability of our CSAM to fully leverage the complementary information within RGB-E data. Furthermore, in the challenges of FOC and OV, where scenes lack discernible appearance information, our CSAM leverages motion information within the scenes to accurately determine target positions. Besides, considering the common occurrence of distractors in tracking scenarios, our desirable performance on various challenges demonstrates the effectiveness of employing MOT philosophy.

## C.2 Visualization of Candidate Features.

In the proposed CSAM framework, differentiating various candidates is the key to improving tracking robustness. In Fig 9, candidate features are visualized via t-SNE. With the help of the proposed spatial encoder, we can better distinguish candidates in the tracking scenes. It demonstrates the effectiveness of motion information provided by the event stream on enhancing tracking performance.

## C.3 Qualitative performance

The visual comparisons between our proposed method and the other four state-of-the-art tracking algorithms, including CEUTrack [24], HRCEUTrack [44], OSTrack [35] and KeepTrack [19], are illustrated in Fig. 10. We can observe that our method performs better than other trackers in these complex scenes, such as cases with background cluster, motion blur and low illumination.

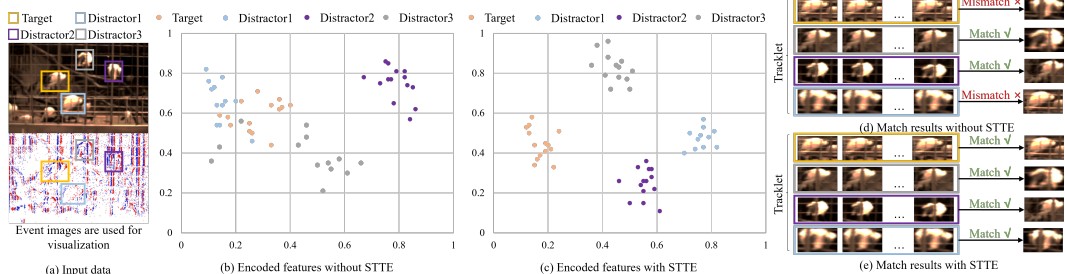

Figure 9: Candidate feature clustered by t-SNE.

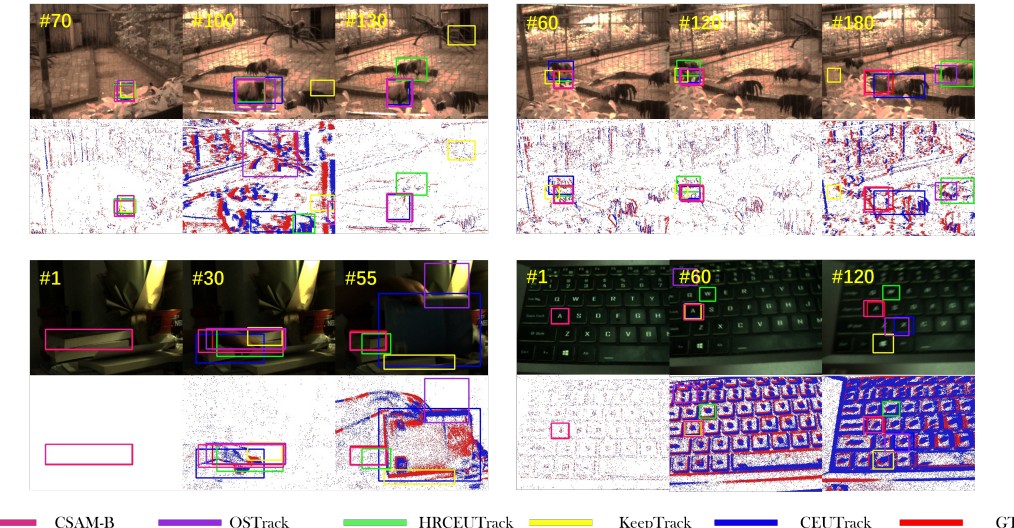

Figure 10: Visualization of tracking results on COESOT dataset. Event images are used for visual comparison only.

### C.4 The impact of CSAM on distractors:

Based on our appearance tracker's response maps, FE108 averages 1.26 distractors per frame, while COESOT has 1.91 distractors per frame. Tracking accuracy declines significantly in the presence of distractors. We categorize sequences into 1) S1: 291 distractor-free sequences and 2) S2: 237 sequences with distractors, as shown in Table 4. Our framework steadily enhances tracking performance in sequences with distractors. What's more, compared with ViPT[42], our tracker achieves notable performance gains of 4.7% and 4.6% in PR and SR, respectively.

### C.5 Experiments of Different MOT Philosophies.

Table 5 presents the performance of using various multi-object tracking methodologies for candidate matching, including KeepTrack [19] and TransSTAM [23]. For fair comparisons, all these variants employ the same appearance model for candidate generation and utilize the same training data and loss functions for model training. The results in Table 5 reveal that our proposed CSAM performs the best. Existing MOT methods generally concentrate on temporal appearance information from the RGB images and motion information from historical candidate locations. In contrast, our proposed CSAM leverages the additional motion information from the event stream, thereby enhancing the robustness of candidate matching.

Table 4: The influence of distractors on COESOT.

| Sequences | ViPT | Appearance Tracker | CSAM | Δ |
|---|---|---|---|---|
| S1 (PR/SR) | 77.6/68.4 | 78.2/68.6 | 78.2/68.6 | - |
| S2 (PR/SR) | 70.8/63.4 | 72.0/62.9 | 75.5/68.0 | 3.5/5.1 |

Table 5: Ablation study of different MOT philosophies.

| Method | KeepTrack [19] | TranSTAM [23] | Ours |
|---|---|---|---|
| PR/SR | 75.1/65.7 | 75.4/66.3 | 76.7/68.1 |

Table 6: Ablation study of T.

| | T=1 | T=5 | T=10 | T=15 | T=20 |
|---|---|---|---|---|---|
| PR | 75.7 | 75.9 | 76.1 | 76.7 | 76.6 |

Table 7: Ablation study of $\tau_{th}$.

| | $\tau_{th}$=0.55 | $\tau_{th}$=0.65 | $\tau_{th}$=0.75 | $\tau_{th}$=0.85 |
|---|---|---|---|---|
| PR | 75.1 | 75.7 | 76.7 | 76.2 |

Table 8: Ablation study of $\zeta$.

| | $\zeta$=0.0 | $\zeta$=0.25 | $\zeta$=0.5 | $\zeta$=0.75 |
|---|---|---|---|---|
| PR | 75.1 | 76.7 | 76.1 | 75.2 |

Table 9: Ablation study of the number of layers in STTE and DBTD.

| Layer Number | 1 | 2 | 3 | 4 |
|---|---|---|---|---|
| PR | 76.7 | 76.9 | 75.7 | 75.3 |

## C.6 Experiments of the hyper-parameters

In the proposed CSAM framework, the involved hyper-parameters are the time interval T, relevance threshold $\tau_{th}$, and classification threshold $\zeta$.

**The ablation study of T:** The ablation study about T is shown in Table 6. When T=1, there is no temporal information included in the proposed framework. T > 1 can improve the association accuracy. In addition, since increasing T also adds more tracklets for the association, it increases the complexity of the association task. Finally, T = 15 is used in all experiments.

**The ablation study of $\tau_{th}$:** We show the ablation studies of $\tau_{th}$ in Table 7, we achieve the best performance when $\tau_{th}$=0.75. When $\tau_{th}$ is too small, the proposed tracker may cause false tracklet-matching. When $\tau_{th}$ is too big, the proposed tracker may not be able to complete the track-matching.

**The ablation study of $\zeta$:** We show the ablation studies of $\zeta$ in Table 8, we achieve the best performance when $\zeta$=0.25 When threshold $\zeta$ is too big, it is hard to relocate the target.

**The number of layers in the proposed STTE and DBTD:** In addition, we provide the experiments about using more layers in STTE and DBTD in Table 8. We found that a layer number of 2 can achieve better performance but at the expense of operational efficiency. Furthermore, more layers will introduce a large number of parameters, which may cause over fitting problems and lead to performance degradation.

## D   Societal impacts

Object tracking has diverse applications extending from visual surveillance systems, autonomous vehicles and intelligent transportation systems. In-depth research on RGB-E tracking has a wide

range of positive impacts on practical applications in these fields. However, the misuse of Object tracking technology can have a negative impact on personal privacy.

