# OpenReview forum: "Revisiting motion information for RGB-Event tracking with MOT philosophy"
_NeurIPS.cc/2024/Conference — NeurIPS 2024 poster_

### Official Review · Reviewer_fErD · 2024-07-12

**Soundness:** 4
**Presentation:** 3
**Contribution:** 3
**Rating:** 6
**Confidence:** 4

**Summary:**

This work proposes a novel RGB-E tracking framework, CSAM. CSAM first predicts candidates and then tracks both targets and distractors using MOT philosophy. Comprehensive experiments are conducted on three RGB-E tracking benchmarks, showing that CSAM achieves state-of-the-art performance.

**Strengths:**

1. The motivation is clear and the problem studied is important
2. This is the first work to introduce MOT philosophy to the SOT task using RGB-E data.
3. The authors proposed a novel RGB-E tracking framework, CSAM.
4. Extensive experimental results back up the effectiveness of CSAM.

**Weaknesses:**

1. Adding more experimental results under varying illumination conditions could be helpful to highlight the necessity of DVS.
2. The paper uses too many abbreviations, especially in CSAM, where each module has its own set of abbreviations, making it hard to remember them all.

**Questions:**

1. Where are the bottlenecks in the inference latency of the CSAM method, and how much time does each module consume? Additionally, the Candidate Embedding Module in CSAM uses NMS, which is likely one of the most time-consuming aspects. If so, can anchor-free methods address this?
2. Why does CSAM-B achieve a high FPS in Table 2 despite having a larger model size and higher FLOPs compared to SeqTrackv2-B256?
3. In MoT, the frequent appearance and disappearance of targets is a major challenge. The applicability of the CSAM method in this context is particularly interesting, as discussed on pages 130-134 of the paper. The authors could include more examples of tracking in situations where objects frequently disappear due to occlusion.
4. The total training time on two RTX 3090 GPUs?
5. Why are RSR and RPR metrics used in FE108, while SR and PR metrics are used in COESOT and VisEvent? What are the distinctions between them?
6. The DVS sensor introduces significant event noise during nighttime or in varying lighting conditions. It appears that the CSAM method implicitly filters out this noise using Transformer. However, the authors voxelized the DVS sensor input without applying denoising, could this potentially degrade performance? Besides, what unique approach does CSAM employ for handling DVS sensor data? CSAM appears to be applicable to other modality combinations.

**Limitations:**

Please see the weaknesses part.

---

> ### Author Rebuttal · Authors · 2024-08-06
>
> We want to express our gratitude for your support and valuable insights. It's reassuring to know that you recognize the originality and efficacy of our research. We would greatly appreciate your continued support in championing our work.
>
> **W1 and Q6. More experimental results under varying illumination conditions**
>
> **Advantages and disadvantages of event stream:**
> As discussed with FE108, VisEvent and COESOT, event cameras (Dynamic Vision Sensors (DVS)) excel at capturing motion information at very low latency,  and are almost free from the trouble of motion blur.  In addition, DVS sensors also outperform the visible (RGB) cameras on dynamic range (140 vs 60dB), which enables them to work effectively even under poor illumination conditions. Simultaneously, we noticed that the event camera cannot provide effective information for tracking of slow-moving or stationary objects.
>
> **Input representation:**
> Previous event trackers usually transform the asynchronous event flows into synchronous event images by stacking the events in a fixed time interval. CEUTrack transforms the event points into
> voxels which preserves the temporal information well, and the voxels top-selection mechanism also fully exploits the sparse representation of event data which reduces the computational complexity. In this paper, we also adopt the same transformation to get the event images but focus on designing a novel tracking framework to explore both the effective appearance information and motion information. Besides, the proposed CSAM framework employs both the appearance information and the motion information to suppress the noisy information within the event data.
>
> **Cannot be directly applied to other multimodal tracking tasks:**
> Considering the motion information provided by the event data, in the proposed STTE, we design a spatial transformer encoder to establish distinctive spatial relationships by using an event stream. In addition, in the proposed DBTD, a motion decoder is introduced to match each tracklet by using the event tokens. Differently, for other multi-modal tracking tasks (e.g., RGB-Thermal, RGB-Depth), it is still difficult for us to construct accurate spatial correlations or match tracklets by using other modality data like thermal images or depth images.
>
> **Attributes Performance:**
> Here, in order to prove the necessity of DVS, we provide the results of with and without event stream under the challenges of low illumination (LI) and Illumination variation (IV). As shown below,  our tracker with event stream shows substantially better results, which demonstrates the effectiveness of DVS.
>
> | Challenges     | LI | IV |
> |---------|------|------|
> | CSAM with event data (PR score)| 74.1 | 82.3|
> | CSAM without event data (PR score)| 50.8 | 71.8|
>
>
> **W2. Too many abbreviations.**
>
> we are very sorry for the confusion. In the revision, we will avoid unnecessary abbreviations, and the major mathematical symbols used in this paper will be summarized in a Table for ease of explanation in the revision.
>
> **Q1. Latency of the CSAM method.**
>
> **Latency of NMS:**
> In this Table, we provide the time each module takes. We found that NMS takes very little time in the PyTorch framework. This is because the tracking scenes usually contain only a limited number of candidates.
> Besides, we can use the cuda version of NMS to speed up its operation.
>
> **Latency of Hungarian algorithm:**
> Similarly, due to the limited number of candidates, the Hungarian algorithm used in multi-target tracking will not take too long. To model the temporal associations in historical frames, the temporal encoder occupies a longer reasoning time in the CSAM framework.
> In the future, we will study how to use a more compact representation of historical information to efficiently establish temporal associations.
>
> | Module | Appearance model | NMS | Candidate Encoding Module| Spatial Encoder|
> |---------|------|------|------|------|
> | Latency (ms) | 13.3 | 0.8| 0.5  | 0.8|
>
> | Module | Temporal encoder | Dual-branch Transformer Decoder| Hungarian algorithm|  |
> |---------|------|------|------|------|
> | Latency (ms) | 2.1| 0.7| 0.6  |  |
>
> **Anchor-free:** Our appearance model follows the anchor-free pipeline to generate a series of candidates. The candidate matching network also does not use anchor boxes. We argue that the design of a one-stage framework in the future can improve the tracking efficiency; this will be our follow-on work.
>
> **Q2. The FPS and FLOPs of SeqTrackv2-B256.**
>
> SeqTrack casts visual tracking as a sequence generation task, forecasting object bounding boxes in an autoregressive manner. Such a design requires the gradual generation of tracking results and frequent access to intermediate calculation results, resulting in lower FLOPs and slower inference speed.
>
> **Q3. More examples of appearance and disappearance of targets.**
>
> We will add more examples in the revision. We also provide the tracking performance under the challenges of FOC (target is fully occluded) and OV (target completely leaves the video sequence), we noticed that our framework significantly improves the tracking performance when targets frequently appear and disappear.
>
> | Challenges     | FOC| OV|
> |---------|------|------|
> | CSAM-B| 70.7 | 70.3|
> | Apperance model| 62.6 | 61.3|
>
> **Q4. The total training time.**
>
> Our proposed model is trained in two stages. In the first stage, the appearance model’s training took 12 hours. In the second stage, the proposed model took 7 hours for training.
>
> **Q5. Different metrics in FE108, COESOT and VisEvent**
>
> As in FENet[38], to mitigate small alignment errors, we utilize two widely used metrics, i.e., maximum precision rate (MPR) and maximum success rate (MSR), to evaluate the tracking performance on FE108. Differently, VisEvent and COESOT employ better alignment, they directly use the PR and SR metrics to evaluate different trackers. All evaluations are tested under the official codes.

---

> > ### Comment · Reviewer_fErD · 2024-08-13
> >
> > I appreciate the authors' response to several of my questions. My rating remains the same.

---

> > > ### Author Response · Authors · 2024-08-13
> > > **Re**
> > >
> > > We are delighted that our rebuttal has addressed your concerns. We sincerely appreciate your recognition of our work.

---

### Official Review · Reviewer_SevM · 2024-07-12

**Soundness:** 3
**Presentation:** 3
**Contribution:** 3
**Rating:** 5
**Confidence:** 5

**Summary:**

A novel RGB-E tracking framework with MOT philosophy has been proposed in order to keep track of both targets and distractors to robustly track a single object. It includes a Candidate Encoding Module, a Spatial-Temporal Transformer Encoder and a Dual-branch Transformer Decoder. Within these modules, the authors exploited the appearance information in combination with motion cues to improve the candidate matching and affinities. Experimental validation is comprehensive and results on three RGB-E object tracking benchmarks are state-of-the-art.

**Strengths:**

1. Using the MOT philosophy with spatial-temporal information fusion to solve the similar distractors problem in RGB-E tracking domain is new attempt.

2. The proposed method achieves the new sota RGB-E SOT tracking results.

**Weaknesses:**

1.The ablation study about N/M/T is missing and the selection of hyperparameters(e.g., τth and ζ) lacks discussion.

2.Does the delay caused by the Hungarian algorithm constitute the majority of the additional time compared with the pure appearance-based tracker?

3.Is the training process carried out in a sequential iterative manner? That is to say, every frame will be output a prediction result or just the last frame in a batch. It is best to provide details in the main paper.

4.Have authors tried more layers in STTE and DBTD? Will they lead to continuous improvement in performance?

5.More thorough checks are needed for the paper writing and illustrations, including but not limited to the following parts:
a. The description about ΓEN in Line216 is not consistent with that in Figure 3
b. In Figure 2, “TIR” should be “Event”. Colors of Appearance and Motion feature embeddings are swapped.
c. In Figure 5,  A^{sa} should be  A^s. M^{t-1}should only have N blocks not M.
d. In Line336 and Line338, “encoder” should be “decoder”.
e. “COEST” should be “COESOT”. It has multiple mis-usages across whole paper.
f. In Figure8, “SGTE” should be “STTE”

**Questions:**

See the Weakness and give the corresponding explaination.

---

> ### Author Rebuttal · Authors · 2024-08-06
>
> We extend our heartfelt gratitude for your perceptive insights and your recognition of the unique and meaningful contributions made by our research. Your support is highly valued, and we would be honored if you could serve as an advocate for our work.
>
> **Q1. The ablation study about N/M/T/$\tau_{th}$/$\zeta$:**
>
> **N and M:** N and M denote the number of candidates of the previous and current frame respectively. Therefore, N and M are actually determined by the tracking sequences and are not one of our settings.
> In supplementary C.5, we provide the influence of candidates to illustrate the effectiveness of the proposed framework.
>
> **The ablation study of T:**
> The ablation study about T is shown below. When T=1, there is no temporal information included in the proposed framework. T > 1 can improve the association accuracy. In addition, since increasing T also adds more tracklets for the association, it increases the complexity of the association task. Finally, T = 15 is used in all experiments.
>
> |   | T=1| T=5| T=10| T=15| T=20|
> |---------|------|------|------|------|------|
> | PR score| 75.7 | 75.9 | 76.1  | 76.7| 76.6|
>
> **The ablation study of $\tau_{th}$:**
> We show the ablation studies of $\tau_{th}$ below. As shown in the following table, we achieve the best performance when $\tau_{th}$=0.75.
> When $\tau_{th}$ is too small, the proposed tracker may cause false tracklet-matching. When $\tau_{th}$ is too big, the proposed tracker may not be able to complete the tracklet-matching.
>
> |   | $\tau_{th}$=0.55| $\tau_{th}$=0.65| $\tau_{th}$=0.75| $\tau_{th}$=0.85|
> |---------|------|------|------|------|
> | PR score| 75.1 | 75.7 | 76.7  | 76.2|
>
> **The ablation study of $\zeta$:**
> We show the ablation studies of $\zeta$ below. As shown in the following table, we achieve the best performance when $\zeta$=0.25
> When threshold $\zeta$ is too big, it is hard to relocate the target.
>
> |   | $\zeta$=0.0| $\zeta$=0.25| $\zeta$=0.5| $\zeta$=0.75|
> |---------|------|------|------|------|
> | PR score| 75.1 | 76.7 | 76.1  | 75.5|
>
> **Q2. The delay of the Hungarian algorithm.**
>
> **Hungarian algorithm:** As shown in the table below, the Hungarian algorithm takes 0.6 ms for object association. Because when we tested on the SOT dataset, most tracking scenes only contained a limited number of candidates (Please refer to Appendix C.5).
>
> **Other parts:** To model the temporal associations in historical frames, the temporal encoder takes a longer reasoning time in the CSAM framework. In the future, we will study how to use a more compact representation of historical information to efficiently establish temporal associations.
>
> | Module | Appearance model | NMS | Candidate Encoding Module| Spatial Encoder|
> |---------|------|------|------|------|
> | Latency (ms) | 13.3 | 0.8| 0.5  | 0.8|
>
> | Module | Temporal encoder | Dual-branch Transformer Decoder| Hungarian algorithm|  |
> |---------|------|------|------|------|
> | Latency (ms) | 2.1| 0.7| 0.6  |  |
>
> **Q3. Training process.**
>
> We do not use the sequential iterative manner during training.
> We employ a similar training strategy as that in KeepTrack, which has been carefully introduced in Appendix A and B.
> For partial supervision, we use two consecutive frames as the historical frames and select one frame to output the prediction results and calculate the training loss.
> For self-supervision training, we only require a single frame and its candidates for training loss calculation.
> We will add relevant content in the main text in the revision.
>
> **Q4. More layers in STTE and DBTD**
>
> We provide the experiments about using more layers in STTE and DBTD in the following Table.
> We found that a layer number of 2 can achieve better performance but at the expense of operational efficiency.
> Furthermore, more layers will introduce a large number of parameters, which may cause overfitting problems and lead to performance degradation.
>
> | Layer Number | 1| 2| 3| 4|
> |---------|------|------|------|------|
> | PR score| 76.7 | 76.9 | 75.7  | 75.3|
>
> **Q5. More thorough checks are needed.**
>
> We are very sorry for the above mistakes in the previous manuscript. We will modify all of these typos in the revision.

---

### Official Review · Reviewer_KN3J · 2024-07-13

**Soundness:** 3
**Presentation:** 3
**Contribution:** 2
**Rating:** 5
**Confidence:** 3

**Summary:**

•	This paper proposes a novel RGB-E tracking framework, i.e., CSAM, which first predicts the candidates by using an appearance model and then keeps track of both targets and distractors with an MOT philosophy. The model show significantly improved state-of-the-art results.

**Strengths:**

1. The paper is well-written and easy to understand.
2. Comparisons over several datasets demonstrate the effectiveness of tracking multiple candidates for robust tracking.

**Weaknesses:**

Because I am not a researcher on RGB-E tracking, and the authors have given me enough information in the main text and appendix, it is difficult for me to point out the shortcomings from a technical perspective. One small suggestion is that the authors gave some examples of qualitative analysis in the supplementary materials, but it is recommended to put some of the content in the main text, so that readers can understand the advantages and disadvantages of the method by combining quantitative numerical results and qualitative examples.

**Questions:**

•	Please refer to weaknesses.

**Limitations:**

Yes

---

> ### Author Rebuttal · Authors · 2024-08-06
>
> We greatly appreciate your constructive and insightful comments. Thanks for your recognition of our work.
>
> **Q1. Putting some qualitative analysis in the main text.**
>
> Due to the limited page space, our previous manuscript only provides the qualitative analysis in the supplementary materials. We will put some of them in the main text in the revision space permitting.
>
> In addition, please allow me to restate the novelty of our work briefly:
>
> 1) To the best of our knowledge, we are the first to introduce the MOT philosophy for the SOT task using RGB-E data. The proposed CSAM framework significantly improves the tracker's ability to cope with tracking drift caused by distractors.
>
> 2) We propose three effective modules: a Candidate Encoding Module, a Spatial-Temporal Transformer Encoder and a Dual-branch Transformer Decoder.
> The Candidate Encoding Module initially aggregates the RGB features and corresponding classification scores to generate the appearance embedding. Meanwhile, the event features and corresponding bounding box coordinates are fused to obtain motion information.
> In the Spatial-Temporal Transformer Encoder, the spatial encoder construct
> spatial correlations of each frame by using motion information, and the temporal encoder establishes the temporal relationships of each tracklet.
> In the proposed Dual-branch Transformer Decoder, we also generate the assignment matrix by using both the appearance information and motion information, simultaneously.
>
> 3) We show significantly improved state-of-the-art results of our proposed method on multiple RGB-E tracking benchmarks.

---

> > ### Comment · Reviewer_KN3J · 2024-08-10
> > **Official Comment by Reviewer KN3J**
> >
> > As I said in my initial comments, I am not particularly familiar with the details of this field. But I am very grateful for the author's response to me. I also read the authors' responses to other reviewers and thanked them for their efforts during the rebuttal. I want to maintain the original score first and refer to other reviewers' opinions on the author's rebuttal. If most reviewers approve of the author's efforts, I think I will consider giving a more positive score.

---

> > > ### Author Response · Authors · 2024-08-12
> > > **Re:**
> > >
> > > Once again, we sincerely thank you for your valuable feedback.
> > >
> > > Best regards

---

### Official Review · Reviewer_fGZm · 2024-07-16

**Soundness:** 2
**Presentation:** 1
**Contribution:** 2
**Rating:** 5
**Confidence:** 5

**Summary:**

This paper focuses on RGB-Event tracking. The authors propose to leverage MOT philosophy to distinguish the distractors to enhance the robustness of the tracker. Following a tracking-by-detection framework, the authors first generate a series of candidates and then match them with historical known priors with a CEM, STTE and DBTD module. The authors conduct experiments on several datasets to demonstrate the proposed methods.

**Strengths:**

- The proposed method achieves state-of-the-art results on several datasets.

**Weaknesses:**

- The motivation of the article is unclear. The article aims to enhance the robustness of the tracker by distinguishing between interfering objects. However, instead of analyzing in detail the causes of the model's inability to distinguish between interfering objects, the authors introduce a number of complex modules to perform the matching. In addition, the authors only analyze the effectiveness of the proposed modules through some experimental results, without detailed and deep analysis such as visually demonstrating the proposed modules through feature visualization, trajectory visualization, and so on. Therefore, I am concerned about the innovativeness of this paper.
- The proposed method seems to be a combination of existing techniques. Spatio-temporal encoder-decoder is widely explored in SOT and MOT community. The authors seem to have only modified some blocks, yet what does that have to do with what the article is claiming to prove.
- The article writing needs to be improved. Line 5-6 is a misrepresentation, beacuse the RGB-Event tracker, which tracks a single target, does not need to track interfering objects. I can understand that the author utilizes the MOT idea, but this representation may misleading the readers.
- The model is complex, and I'm concerned about whether the model is far more complex and computationally intensive than other algorithms.
- No open source code

**Questions:**

Spatio-temporal architecture is widely used in SOT and MOT, what is the novelty of the article?

**Limitations:**

Please refer to the above weaknesses.

---

> ### Author Rebuttal · Authors · 2024-08-06
>
> We greatly appreciate your constructive and insightful comments. Thanks for your recognition of our work.
>
> **Q1. The motivation of the paper is unclear. Lack of analysis.**
>
> **Motivation:**
> As shown in Fig. 1 (a) in the manuscript, the co-occurrence of distractor objects similar in appearance to the target is a common problem in the SOT task. Most SOT trackers based on appearance information struggle to identify the target in such cases, often leading to tracking failure.
>
> Existing RGB-E trackers primarily concentrate on exploring complementary appearance information within RGB and event data to enhance tracking performance.
> Despite achieving commendable improvements, mainstream RGB-E tracking algorithms still cannot solve the association problem of the targets and distractor objects in the temporal domain.
>
> Some RGB trackers (e.g., KYS, KeepTrack, TrDiMP and DMTrack) propagate valuable scene information to improve the discriminative ability.
> However, these methods are susceptible to environmental interference, and their matching strategies relying on appearance information may miss the target when the target and distractor trajectories are close.
>
> In fact, event data can not only provide the edge information to improve the RGB feature representations but also contains abundant motion cues to reflect the motion state of the objects, which is meaningful to differentiate between targets and distractors, even if they may look similar. The event camera offers a new perspective to reflect the motion state of the objects, which is crucial to accurately match the tracklet in the tracking sequences.
>
> Therefore, we propose an Appearance-Motion Modeling RGB-E tracking framework to dynamically incorporate motion information as well as appearance information contained in the RGB-E videos to track all the candidates with a Multi-Object Tracking (MOT) philosophy, thus avoiding the tracking drift.
>
> **Experimental results:**
> We provide the qualitative analysis (trajectory visualization) in lines 634-644 (Fig 9， Fig 10).
> In addition, we provide the visualization of candidate features via t-SNE in lines 629-633 (Fig 8).
> Furthermore, more ablation studies are covered in Appendix C to verify the effectiveness of the proposed framework. We will add more visualization results in the revision.
>
>
> **Q2. Spatio-temporal encoder-decoder is explored in SOT and MOT community**
>
> **Different Motivations with existing SOT methods:**
> Some SOT works (e.g., TrDiMP, TCTrack, HIPTrack) employ the encoder-decoder structure to explore historical information.
> Most of these methods do not explicitly distinguish the motion trajectories of distractors in the scene.
> The encoder and decoder are usually used to combine historical features and enhance current frame features, respectively.
> Our work is different, in the proposed CSAM framework, the encoder establishes the association between different candidates in each frame and the temporal relationships of each candidate. And the decoder is used to match candidates with historical tracklets.
>
> **Different Design with existing  MOT methods:**
> Numerous MOT methods use the encoder-decoder structure to construct the temporal relationships and generate the assignment matrix for matching. However, most of them rely on the appearance information from the input RGB data.
> These methods are susceptible to environmental interference, and their matching strategies relying on appearance information may miss the target when the target and distractor trajectories are close.
> Differently, in the proposed CSAM framework, taking into account the fact that event data can provide reliable motion status, we fully exploit the appearance and motion information in the tracking scene during the encoding and decoding stages to enhance tracking performance.
>
>
> **Q3. Improving Writing.**
>
> Sorry for the unclear description. We will modify the abstract in the revision as follows: RGB-Event single object tracking (SOT) aims to leverage the merits of RGB and event data to achieve higher performance.
> However, existing frameworks focus on exploring complementary appearance information within multi-modal data, and struggle to address the association problem of targets and distractors in the temporal domain using motion information from the event stream.
> In this paper, we introduce the Multi-Object Tracking (MOT) philosophy into RGB-E SOT to keep track of targets as well as distractors by using both RGB and event data.
>
> **Q4. The model is complex.**
>
> In lines 318-324, Table 2 shows that our CSAM introduces limited computation costs (5.6 ms latency and 14.4M parameters), while significantly improving the tracking performance.
> In addition, compared with recently advanced trackers, i.e., ViPT and SeqTrackv2-L256, our model size has not increased significantly and we can still run at real-time speeds.
>
> **Q5: No open source code.**
>
> We describe the proposed methods in Section 3 and provide the implementation details in Section 4.1, Appendices A and B.
> We illustrate the exact command and environment needed to run to reproduce
> the results. We will release the source code when the paper is accepted.

---

> > ### Comment · Reviewer_fGZm · 2024-08-11
> >
> > Thank the authors for their reply. I have updated my given score.

---

> > > ### Author Response · Authors · 2024-08-12
> > > **Re**
> > >
> > > We greatly appreciate your recognition of our work and the increased score!

---

### Author Rebuttal · Authors · 2024-08-06

We sincerely appreciate the comprehensive reviews provided by all the reviewers. The valuable feedback has significantly contributed to improving the quality of our manuscript. We extend our gratitude to Reviewer KN3J, Reviewer SevM, and Reviewer fErD for recognizing the novelty of our work. Their positive acknowledgments of our research's innovation are greatly appreciated. Additionally, we kindly request Reviewer fGZm to reconsider our work after reviewing our response. Your reconsideration will be highly valued.

Based on the comments from the reviewers, I have summarized the strengths of our paper as follows:

1. The utilization of event data combined with the MOT philosophy can ensure more robust target identification for SOT tasks.

2. The proposed CSAM framework only introduces limited parameters and computational costs but significantly improves the tracking performance.

3. The proposed method achieves very competitive performance compared with state-of-the-art methods. Also, an ablation study is conducted to validate the effectiveness of different components in the proposed method.

4. There is currently a lack of research on multi-modal multi-target tracking. This paper can provide a new perspective for future research.

We have summarized our novelty as follows:

1. To the best of our knowledge, we are the first to introduce the MOT philosophy for the SOT task using RGB-E data. The proposed CSAM framework significantly improves the tracker's ability to cope with tracking drift caused by distractors.

2. We propose three effective modules: a Candidate Encoding Module, a Spatial-Temporal Transformer Encoder and a Dual-branch Transformer Decoder.
The Candidate Encoding Module initially aggregates the RGB features and corresponding classification scores to generate the appearance embedding. The event features and corresponding bounding box coordinates are also fused to obtain motion information.
In the Spatial-Temporal Transformer Encoder, the spatial encoder constructs
spatial correlations of each frame by using motion information, and the temporal encoder establishes the temporal relationships of each tracklet.
In the proposed Dual-branch Transformer Decoder, we also generate the assignment matrix by using both the appearance information and motion information, simultaneously.

3. We show significantly improved state-of-the-art results of our proposed method on multiple RGB-E tracking benchmarks.

We believe that these innovative contributions enhance the value and significance of our research in the field of object tracking. We kindly request the reviewers to reassess our work in light of these contributions and extend their support to our efforts.

We plan to implement the reviewers' insightful suggestions by incorporating additional essential experiments. Furthermore, we will thoroughly review the manuscript to correct any typographical and grammatical errors.

We have provided detailed responses to each reviewer, carefully addressing all specific points raised. We sincerely thank the reviewers for their valuable feedback and appreciate the dedication of the program chair and area chair. Your support in our endeavors is greatly appreciated. We are fully committed to addressing the concerns raised and refining our manuscript accordingly.

---

### Decision · Program_Chairs · 2024-09-25

**Decision:**

Accept (poster)

**Comment:**

This paper proposes a novel framework that integrates Multi-Object Tracking (MOT) philosophy into RGB-Event (RGB-E) tracking, aiming to enhance robustness by distinguishing between targets and distractors. The approach utilizes a Candidate Encoding Module, a Spatial-Temporal Transformer Encoder, and a Dual-Branch Transformer Decoder to effectively leverage both appearance and motion information. Some reviewers raised concerns about the clarity of the motivation and the complexity of the proposed model. Suggestions were made to include more qualitative analysis and refine the writing for better clarity. Despite these concerns, the paper presents a valuable contribution to the field of RGB-E tracking, especially by addressing the challenges of distractors. The ACs recommend accepting the paper, with the expectation that the authors will address the reviewers' feedback in the final version.